# Clinical and biological clusters of sepsis patients using hierarchical clustering

**Grégory Papin**[1,2]*, **Sébastien Bailly**[3,4], **Claire Dupuis**[1,2], **Stéphane Ruckly**[5], **Marc Gainnier**[6], **Laurent Argaud**[7], **Elie Azoulay**[8], **Christophe Adrie**[9], **Bertrand Souweine**[10], **Dany Goldgran-Toledano**[11], **Guillaume Marcotte**[12], **Antoine Gros**[13], **Jean Reignier**[14], **Bruno Mourvillier**[15], **Jean-Marie Forel**[16], **Romain Sonneville**[2], **Anne-Sylvie Dumenil**[17], **Michael Darmon**[8], **Maité Garrouste-Orgeas**[18], **Carole Schwebel**[19], **Jean-François Timsit**[1,2], **OUTCOMEREA study group**[¶]

**1** UMR 1137 IAME INSERM- Paris Diderot University, Paris, France, **2** Medical and Infectious Diseases ICU, Bichat University Hospital, AP-HP, Paris, France, **3** HP2 Laboratory, INSERM U1042, Univ. Grenoble Alpes, Grenoble, France, **4** EFCR laboratory, Grenoble Alpes University Hospital, Grenoble, France, **5** Biostatistics Department, OUTCOMEREA, Bobigny, France, **6** Medical ICU, La Timone University Hospital, Marseille, France, **7** Medical ICU, Edouard Herriot University Hospital, Lyon, France, **8** Medical ICU, Saint Louis University Hospital, AP-HP, Paris, France, **9** Polyvalent ICU, Delafontaine Hospital, Saint-Denis, France, **10** Medical ICU, Gabriel Montpied University Hospital, Clermont-Ferrand, France, **11** Medical ICU, Le Raincy-Montfermeil Hospital, Montfermeil, France, **12** Surgical ICU, Edouard Herriot University Hospital, Lyon, France, **13** Medical-Surgical ICU, Versailles Hospital, Versailles, France, **14** Medical ICU, Nantes University Hospital, Nantes, France, **15** Medical Intensive Care Unit, Robert Debré University Hospital, Reims, France, **16** Medical ICU, Hôpital Nord University Hospital, Marseille, France, **17** Surgical ICU, Antoine Béclère University Hospital, AP-HP, Clamart, France, **18** Medical Unit, French-British Hospital Institute Levallois-Perret, Levallois-Perret, France, **19** Medical ICU, Grenoble 1 University, Albert Michallon Hospital, Grenoble, France

¶ The manuscript was written on behalf of the OUTCOMEREA study group listed in the acknowledgements.
* grego.papin@gmail.com

**Data Availability Statement:** All relevant data are within the manuscript and its Supporting Information files.

## Abstract

### Background

Heterogeneity in sepsis expression is multidimensional, including highly disparate data such as the underlying disorders, infection source, causative micro-organismsand organ failures. The aim of the study is to identify clusters of patients based on clinical and biological characteristic available at patients' admission.

### Methods

All patients included in a national prospective multicenter ICU cohort OUTCOMEREA and admitted for sepsis or septic shock (Sepsis 3.0 definition) were retrospectively analyzed. A hierarchical clustering was performed in a training set of patients to build clusters based on a comprehensive set of clinical and biological characteristics available at ICU admission. Clusters were described, and the 28-day, 90-day, and one-year mortality were compared with log-rank rates. Risks of mortality were also compared after adjustment on SOFA score and year of ICU admission.

### Results

Of the 6,046 patients with sepsis in the cohort, 4,050 (67%) were randomly allocated to the training set. Six distinct clusters were identified: young patients without any comorbidities,

**Funding:** This study received partial support from MIAI @ Grenoble Alpes (ANR-19-P3IA-0003), awarded to SB. No additional external funding was received for this study.

**Competing interests:** The authors have declared that no competing interests exist.

admitted in ICU for community-acquired pneumonia (n = 1,603 (40%)); young patients without any comorbidities, admitted in ICU for meningitis or encephalitis (n = 149 (4%)); elderly patients with COPD, admitted in ICU for bronchial infection with few organ failures (n = 243 (6%)); elderly patients, with several comorbidities and organ failures (n = 1,094 (27%)); patients admitted after surgery, with a nosocomial infection (n = 623 (15%)); young patients with immunosuppressive conditions (e.g., AIDS, chronic steroid therapy or hematological malignancy) (n = 338 (8%)). Clusters differed significantly in early or late mortality (p < .001), even after adjustment on severity of organ dysfunctions (SOFA) and year of ICU admission.

## Conclusions

Clinical and biological features commonly available at ICU admission of patients with sepsis or septic shock enabled to set up six clusters of patients, with very distinct outcomes. Considering these clusters may improve the care management and the homogeneity of patients in future studies.

## Introduction

In European intensive care units (ICU), the frequency of admission for sepsis still ranges from 10 to 64% [1]. Although the prognosis has improved thanks to a better management of vital organ support, the in-ICU mortality of patients with sepsis still ranges between 20 to 30% [2]. Despite many studies, the lack of effective specific therapies remains the main issue for the management of septic patients. The heterogeneity of patients included in studies focusing on sepsis could partly explain these failures [3–5]. Sepsis is not a specific illness, but rather a syndrome encompassing a still-uncertain pathobiology. Currently, the classification of sepsis is only based on the etiology, the distinction between sepsis and septic shock, and mortality risk levels after stratification on severity scores [6]. These scores do not capture adequately the heterogeneity of sepsis [7]. As explained by J. Castela Forte, to personalize and improve treatments of sepsis, patients must be clustered into common phenotypes based on clinically objective parameters reflecting disease mechanisms. After external validation step, these clusters could be grouped by underlying causal mechanisms and will improve patient characterization; optimized design and powering of randomized control trials. Finally, these clusters can allow identifying differential response patterns by considering baseline characteristics of sepsis patients [8]. Cluster analysis refers to statistical methods of data partitioning whereby objects or individuals are grouped into homogeneous groups on the basis of similarity independent of any outcome variables. Several methods may be used, each of them with advantage and pitfalls [9]. The multiple correspondence analysis (MCA) approach, combined with hierarchical clustering (HC) has already been used in several diseases such as chronic obstructive pulmonary disease, asthma, obstructive sleep apnea, lung cancer and vasculitis [10–14]. This clustering method has two advantages; it limits the noise in the data set without inducing bias by reducing the variables in coordinates obtained for each patient which summarize the main part of the information and does not require defining *a priori* the number of clusters.

In ICU, acute respiratory distress syndrome (ARDS) was evaluated through a cluster analysis [15]. Two clusters were described, with consistent outcomes and responses to treatment; these two ARDS clusters were a tangible discovery made thanks to a retrospective analysis of negative clinical trials. In sepsis, several studies used cluster analysis mainly for genotypic or

transcriptomic approach [16,17]. Recently, Seymour *et al* conducted a clustering analysis on several large cohorts of patients meeting the definition of sepsis 3.0 [18]. Four interesting patient clusters were identified that correlated with host-response patterns and clinical outcomes. However, understanding the phenotype of critically ill patients should include more data characterizing patient comorbidity and infection [19]. The comparison of cluster analysis on different cohorts is therefore essential.

The primary objective of our study was to identify clusters among patients with sepsis by considering data available at admission including: underlying disorders, source of infection, micro-organism, biological host response and organ failures. The secondary objectives were to assess their heterogeneity on outcomes and validate them in an independent dataset.

## Materials and methods

### Study design and data source

We conducted a retrospective analysis of a prospective observational multicenter database (OutcomeRea™). The database, fed by 20 French ICUs, collects prospective data on daily disease severity, iatrogenic events, and nosocomial infections. Each year, each ICU includes a random sample of at least 50 patients who have ICU stays longer than 24 h. Each ICU could choose to obtain the random sample by taking either consecutive admissions to selected ICU beds throughout the year or consecutive admissions to all ICU beds for 1 month. This study was approved by our institutional review board (CECIC Clermont-Ferrand—IRB n˚5891; Ref: 2007–16), which waived the need for signed informed consent of the participants, in accordance with French legislation on non-interventional studies. However, the patients and their next of kin were asked whether they were willing to participate in the database and use of their personal anonymized data, none declined participation.

### Participants

All patients admitted in ICU for sepsis and septic shock were included. ICU-acquired infections were excluded. The presence or absence of infection at admission was prospectively recorded by clinical physicians according to the standard definitions developed by the Centers for Disease Control and Prevention and recently updated [21].

According to the new Sepsis 3.0 definition [5], sepsis was defined as a life-threatening organ dysfunction, identified by an increase by 2 points or more of the SOFA score, associated with an infection. Patients were included in the OUTCOMEREA™ database prior to this new definition. Accordingly, the increase in the SOFA score was retrospectively calculated. The maximum SOFA score measured on the first day of ICU stay was used. Prior to admission in ICU, SOFA score baseline can be considered to be at zero. The baseline SOFA score of patients with chronic renal replacement therapy was assumed at 4 points. Septic shock was defined by a need to administer vasopressor agents for maintaining a mean arterial pressure of 65 mmHg or greater, and a serum lactate level greater than 2 mmol/L (>18 mg/dL) in the absence of hypovolemia.

### Data collection

Data were collected daily in the participating ICUs by senior physicians and/or specifically trained study monitors. For each patient, the investigators entered the data into a computer case-report form using data-capture software (RHEA; OutcomeRea™, France) and imported all records into the OutcomeRea™ database. All codes and definitions were established prior to study initiation. For most of the study variables, the data-capture software immediately ran an

automatic check for internal consistency, generating queries that were sent to the ICUs for resolution before incorporation of the new data into the database. In each participating ICU, data quality is checked by having a senior physician from another participating ICU who performs a review of a 2% random sample of the study data every other year. A 1-day data-capture training course held once a year is open to all OutcomeRea™ investigators and study monitors. All qualitative variables used in the analyses had Cohen's kappa coefficient > 0.8 and all variables had inter-rater coefficients in the 0.67 to 1 range, indicating good to excellent reproducibility.

## Statistical methods

For variables with less than 20% of missing values, we performed a multiple imputation of missing data using Markov Chain Monte Carlo. Sixty-three clinical and biological variables were available at admission. Description of 63 variables included in the cluster analysis was available S1 Table. The original data set was randomly split into a training set (2/3 of the patients) and a validation set (1/3 of the patients). The statistical analyses comprised 4 steps: 1) Reduction dimension and cluster analysis; 2) Cluster description, 3) Outcomes, 4) Binary tree and cluster validation (Fig 1). Due to the absence of recommendations, we empirically chose the combination of MCA and HC.

**Cluster analysis.** An MCA was performed to reduce the dimension of the 63 variable of the dataset (**S1 Table**) in "Euclidian patient-coordinates" dataset. Because MCA is based on qualitative variables, quantitative variables we categorized. The first 52 dimensions out of a total of 79, which explained at least 90% of the total variability, were considered in the HC [20]. The HC was performed on this patient-coordinates dataset using the Ward's minimum-variance. Initially, each patient was his own cluster, and was thereafter merged into larger clusters to minimize the within-cluster homogeneity and to maximize the inter-cluster heterogeneity. The final number of clusters was defined on the basis of the Semi partial R-Squared, the Squared-R, Pseudo F statistic and The Pseudo $t^2$ statistic. There is no consensus in the

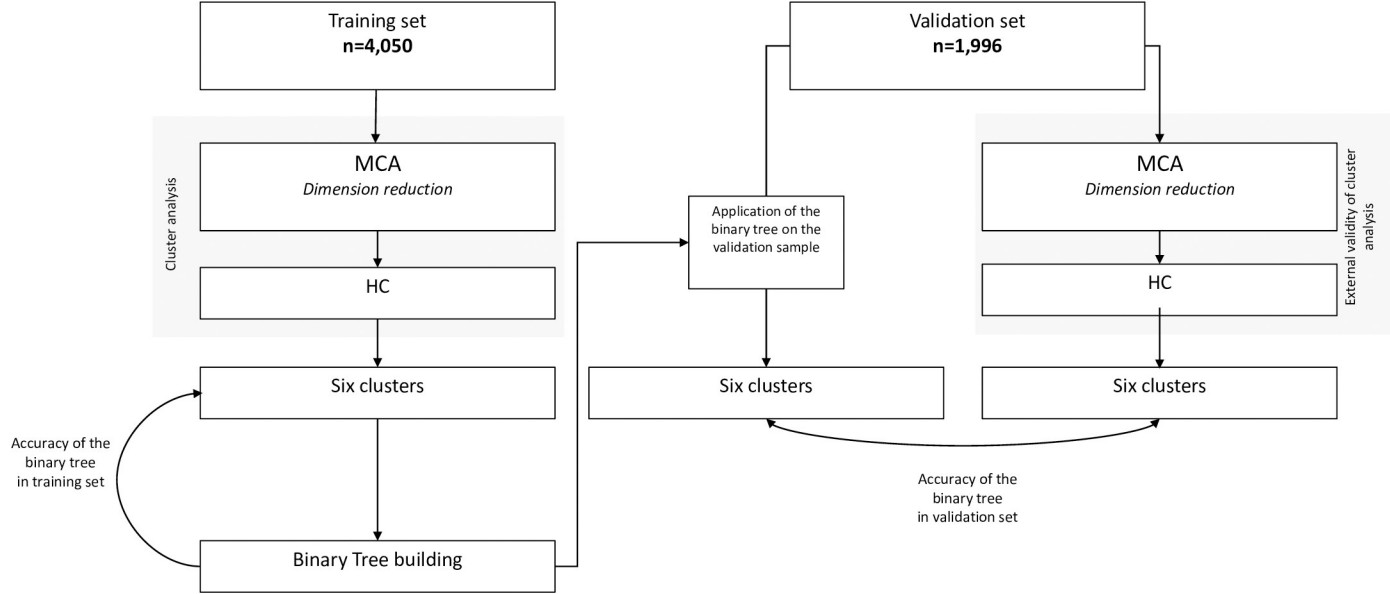

**Fig 1. Schematic of study.** Definition of abbreviations: MCA = multiple correspondence analysis; HC = hierarchical clustering Sensitivity analyses: Cluster analysis in the training set after excluding COPD exacerbation and cluster analysis in the training set after excluding the data before 2008.

literature on the final choice of the number of clusters regardless of the clustering method. K-means algorithm was considered to identify potential outliers in the dataset [21].

**Cluster description.**   Variables were described separately for each cluster, by the use of median and interquartile range (IQR) for quantitative variables, and frequency and percent for qualitative variables. The probability for each variable of belonging to one cluster was assessed using odds ratios, determined by a univariable logistic regression.

**Outcomes.**   The associations between clusters and mortality were assessed by using the status of the patients at day-28, day-90 and one year after admission. The log-rank test was used to compare clusters for mortality. Risk of mortality were described after adjustment on SOFA score at admission using Cox model. Analyses were performed on sub-groups of patients with septic shock. The length of ICU and hospital stay, the number of ventilator-free days at day-28, the duration of the renal replacement therapy, the number of catecholamine-free days at day-28, and the number of organ system failure-free days at day-28 were evaluated.

**Binary tree.**   To build a simple tool able to assign a new patient into clusters using only data that are commonly available at ICU admission, a binary tree was performed using classification and regression tree (CART) [22,23]. The accuracy of the binary tree was evaluated using sensitivity, specificity and Area under the Receiver Operating Characteristic (AUC) in the training set.

**Cluster validation.**   Cluster analysis was also applied in the validation set. Results obtained were compared to clusters obtained by applying the binary tree using AUC. New cluster analysis and cluster description were performed after exclusion of the oldest data (patients admitted before 2008).

Data analyses were performed using R (The R foundation, Vienna, Austria) and SAS version 9.4 (SAS Institute Inc., Cary, NC) [24].

## Results

### Subject demographics

The new definition of sepsis applied on the 18,840 patients of the OUTCOMEREA® database yielded a total of 6,046 patients admitted for a first episode of sepsis between 1997 and 2015, of whom 58% (n = 3,479) had a septic shock (S1 Fig). Details of missing data are available (S1 Table). The initial sample was split in two sets: a training set (n = 4,050 (67%)) and a validation set (n = 1,996 (33%)) (Fig 1 and S1 Table).

In the training set, the median age of patients was 65 year (IQR: [53–76]) and 62% (n = 3,763) were males. Patients were mainly admitted for medical reasons (80%). Median SAPSII and SOFA score at admission were 46 [34–60] and 6 [4–9], respectively. At admission, 58% of patients (n = 2,344) required mechanical ventilation, 60% (n = 2,430) required vasopressor therapy and 10% (n = 415) required renal replacement therapy. The estimated mortality was 26% [95%CI:24–27] at day-28, 36% [95%CI:34–38] at day-90 and 40% [95%CI:38–42] at one year. The baseline characteristics of all variables of the training set are shown in Tables 1–4.

### Clusters analysis and clusters description

Six clusters were identified (S2 and S3 Figs). The contributions of variables in the construction of the first four dimensions of the MCA are depicted on S4 Fig. Representation of patients in the first four dimensions of MCA is depicted on S5 Fig. Clusters were described according to their major associations (S6 Fig). Tables 1–4 describe the distribution of demographic characteristics, various comorbidities, sources of infection, micro-organisms, clinical and biological

**Table 1. Host characteristics by cluster (performed in training set).**

| Variable | Cluster 1 n = 1,603 | Cluster 2 n = 149 | Cluster 3 n = 243 | Cluster 4 n = 1,094 | Cluster 5 n = 623 | Cluster 6 n = 338 |
|---|---|---|---|---|---|---|
| Age (years) | 59 [47–72] | 55 [36–66] | 70 [61–77] | 72 [62–79] | 68 [56–77] | 56 [42–64] |
| Sex (Male) | 1062 (66%) | 81 (54%) | 144 (59%) | 624 (57%) | 366 (59%) | 214 (63%) |
| Weight (kg) | 69 [57–80] | 68.6 [58–80] | 68.6 [58–83] | 70.7 [60–83.5] | 72 [61–84] | 70 [60–80] |
| Malnutrition | 96 (6%) | 5 (3%) | 10 (4%) | 66 (6%) | 33 (5%) | 20 (6%) |
| Alcohol abuse | 302 (19%) | 13 (9%) | 39 (16%) | 171 (16%) | 79 (13%) | 21 (6%) |
| Not complicated diabetes | 131 (8%) | 15 (10%) | 32 (13%) | 222 (20%) | 69 (11%) | 22 (7%) |
| Complicated diabetes | 20 (1%) | 4 (3%) | 10 (4%) | 147 (13%) | 29 (5%) | 3 (1%) |
| Chronic heart failure | 135 (8%) | 10 (7%) | 47 (19%) | 403 (37%) | 123 (20%) | 31 (9%) |
| Chronic kidney disease | 38 (2%) | 6 (4%) | 7 (3%) | 201 (18%) | 37 (6%) | 26 (8%) |
| Liver cirrhosis | 75 (5%) | 9 (6%) | 4 (2%) | 178 (16%) | 65 (10%) | 21 (6%) |
| COPD | 371 (23%) | 9 (6%) | 202 (83%) | 217 (20%) | 68 (11%) | 25 (7%) |
| Hematological malignancy | 41 (3%) | 7 (5%) | 3 (1%) | 29 (3%) | 14 (2%) | 293 (87%) |
| HIV/AIDS or Transplant | 97 (6%) | 20 (13%) | 0 (0%) | 43 (4%) | 14 (2%) | 57 (17%) |
| Solid tumor | 138 (9%) | 7 (5%) | 12 (5%) | 181 (17%) | 129 (21%) | 53 (16%) |
| Chronic steroid therapy | 92 (6%) | 11 (7%) | 18 (7%) | 63 (6%) | 25 (4%) | 41 (12%) |
| Charlson score | 2 [1–3] | 1 [1–3] | 3 [2–4] | 4 [3–6] | 3 [3–5] | 3 [2–4] |
| *ICU Admission* | | | | | | |
| Medical admission | 1494 (93%) | 144 (97%) | 237 (98%) | 957 (87%) | 76 (12%) | 331 (98%) |
| Unscheduled surgery | 46 (3%) | 3 (2%) | 2 (1%) | 89 (8%) | 489 (78%) | 7 (2%) |
| Scheduled surgery | 63 (4%) | 2 (1%) | 4 (2%) | 48 (4%) | 58 (9%) | 0 (0%) |

Definition of abbreviations: COPD = chronic obstructive pulmonary disease; HIV = human immunodeficiency virus; AIDS = acquired immune deficiency syndrome; IQR = interquartile range; **Cluster 1** = young patients without any comorbidities, admitted in ICU for community-acquired pneumonia; **Cluster 2** = young patients without any comorbidities, admitted in ICU for meningitis or encephalitis; **Cluster 3** = elderly patients with COPD, admitted in ICU for bronchial infection with few organ failures; **Cluster 4** = elderly patients, with several comorbidities and organ failures; **Cluster 5** = patients admitted after surgery with a nosocomial infection; **Cluster 6** = young patients, with immunosuppressive disease or therapy, such as AIDS, chronic steroid therapy or hematological malignancy. Values in Numbers (%) or median [IQR].

data, and the organ failures at admission. The main characteristics of each cluster are described below.

**Cluster 1 (1,603 patients, 40%): Young patients without any comorbidities, admitted in ICU for community-acquired pneumonia.** Most patients were included in this cluster. Among them, 1,391 (87%) had pneumonia, which was community-acquired for 1,125 (70%) of them. The most frequently involved micro-organism was *Streptococcus pneumoniae* (259 (16%) patients); however, no responsible pathogens were identified in 710 (44%) patients.

**Cluster 2 (149 patients, 4%): Young patients without any comorbidities, admitted in ICU for meningitis or encephalitis.** This cluster, very close to the cluster 1 in the hierarchical classification, was the smallest. It gathered the youngest patients (median age 55.2 [36.6–66.8]). Almost 20 patients (13%) had HIV, AIDS, or an organ transplant. Their Glasgow Coma Score was the lowest (median at 9 [6–13]) compared to that of other clusters.

**Cluster 3 (243 patients, 6%): Elderly patients with COPD, admitted in ICU for bronchial infection and with few organ failures.** With a median SOFA score of 4 [2–6] at admission, this cluster had the less severely ill patients. Only 68 patients (28%) were in septic shock. COPD was the main comorbidity (202 patients, 83%). The pathogen of the bronchial infection was most often not identified (198 patients, 81%).

**Cluster 4 (1,094 patients, 27%): Elderly patients with several comorbidities and organ failures.** This cluster comprised the oldest patients (median age, 72.3 [62.3–79.7] years).

**Table 2. Source of infection and micro-organism by cluster (performed in training set).**

| Variable | Cluster 1 n = 1,603 | Cluster 2 n = 149 | Cluster 3 n = 243 | Cluster 4 n = 1,094 | Cluster 5 n = 623 | Cluster 6 n = 338 |
|---|---|---|---|---|---|---|
| *Source of infection* | | | | | | |
| Pulmonary | 1391 (87%) | 26 (17%) | 1 (0%) | 431 (39%) | 53 (9%) | 171 (51%) |
| Bronchial | 0 (0%) | 0 (0%) | 237 (98%) | 2 (0%) | 1 (0%) | 0 (0%) |
| Urinary tract | 43 (3%) | 1 (1%) | 0 (0%) | 351 (32%) | 16 (3%) | 16 (5%) |
| Surgical abdomen | 3 (0%) | 0 (0%) | 1 (0%) | 14 (1%) | 419 (67%) | 10 (3%) |
| Medical abdomen | 32 (2%) | 0 (0%) | 0 (0%) | 62 (6%) | 16 (3%) | 38 (11%) |
| Soft tissues | 15 (1%) | 1 (1%) | 2 (1%) | 75 (7%) | 66 (11%) | 13 (4%) |
| Meningeal encephalitis | 0 (0%) | 149 (100%) | 0 (0%) | 0 (0%) | 0 (0%) | 1 (0%) |
| Miscellaneous sites | 44 (3%) | 8 (5%) | 2 (1%) | 88 (8%) | 49 (8%) | 26 (8%) |
| Unknown site | 68 (4%) | 1 (1%) | 2 (1%) | 123 (11%) | 13 (2%) | 68 (20%) |
| *Infection micro-organisms* | | | | | | |
| *Escherichia coli* | 58 (4%) | 2 (1%) | 3 (1%) | 352 (32%) | 122 (20%) | 60 (18%) |
| Other *Enterobacteriaceae* | 99 (6%) | 1 (1%) | 3 (1%) | 208 (19%) | 78 (13%) | 24 (7%) |
| *Pseudomonas* spp. and other NF GNB | 86 (5%) | 1 (1%) | 8 (3%) | 97 (9%) | 46 (7%) | 52 (15%) |
| *Streptococcus pneumoniae* | 259 (16%) | 36 (24%) | 3 (1%) | 39 (4%) | 3 (0%) | 11 (3%) |
| *Enterococcus* and *Streptococcus* | 70 (4%) | 5 (3%) | 9 (4%) | 131 (12%) | 149 (24%) | 25 (7%) |
| *Staphylococcus aureus* | 153 (10%) | 9 (6%) | 1 (0%) | 146 (13%) | 45 (7%) | 17 (5%) |
| Fungus | 18 (1%) | 1 (1%) | 0 (0%) | 27 (2%) | 35 (6%) | 19 (6%) |
| Virus | 54 (3%) | 18 (12%) | 2 (1%) | 21 (2%) | 0 (0%) | 19 (6%) |
| Other pathogens | 341 (21%) | 51 (34%) | 21 (9%) | 140 (13%) | 116 (19%) | 69 (20%) |
| Unknown pathogen | 710 (44%) | 49 (33%) | 198 (81%) | 221 (20%) | 122 (20%) | 60 (18%) |
| Bacteriemia | 142 (9%) | 29 (19%) | 3 (1%) | 320 (29%) | 97 (16%) | 81 (24%) |
| Nosocomial | 478 (30%) | 32 (21%) | 43 (18%) | 369 (34%) | 316 (51%) | 147 (43%) |
| MDRO | 110 (7%) | 8 (5%) | 12 (5%) | 139 (13%) | 70 (11%) | 22 (7%) |

*Definition of abbreviations*: NF GNB: non-fermentative Gram negative bacilli. **Cluster 1** = young patients without any comorbidities, admitted in ICU for community-acquired pneumonia; **Cluster 2** = young patients without any co-morbidities, admitted in ICU for meningitis or encephalitis; **Cluster 3** = elderly patients with COPD, admitted in ICU for bronchial infection with few organ failures; **Cluster 4** = elderly patients, with several comorbidities and organ failures; **Cluster 5** = patients admitted after surgery with a nosocomial infection; **Cluster 6** = young patients with immunosuppressive disease or therapy, such as AIDS, chronic steroid therapy or hematological malignancy. Values in Numbers (%) or median [IQR].

Comorbidities were mainly chronic heart failure (n = 403, 37%), mellitus diabetes (n = 369, 34%) and COPD (n = 217, 20%). The patients in this cluster had the highest severity status at admission, as 799 (73%) had a septic shock and a median SOFA score at admission at 8 [5–11].

**Cluster 5 (623 patients, 15%): Patients admitted after surgery (scheduled or unscheduled), and with a nosocomial infection.** This cluster included 489 patients (78%) admitted for unscheduled surgery and 58 patients (9%) admitted for scheduled surgery. Patients were aged 68.8 [56.8–77.6] in median, and had comorbidities such as solid tumor (129 (21%)) and chronic heart failure (123 (20%)). Most patients were in septic shock (470 (75%)), and their infection was nosocomial for 316 (51%) of them.

**Cluster 6 (338 patients, 8%): Young patients with immunosuppressive conditions such as AIDS, chronic steroid therapy or hematological malignancy.** In this cluster, 293 patients (87%) had a hematological malignancy and 57 (17%) were diagnosed with HIV, AIDS or had an organ transplant. Thrombocytopenia (median platelets count: 40,000 [2424,000–80,000]/mm$^3$) and leucopenia (median leukocytes count: 800 [200–4,300]/mm$^3$) were the main biologic disorders.

**Table 3. Clinical and biological data at admission by cluster (performed in training set).**

| Variable | Cluster 1 n = 1,603 | Cluster 2 n = 149 | Cluster 3 n = 243 | Cluster 4 n = 1,094 | Cluster 5 n = 623 | Cluster 6 n = 338 |
|---|---|---|---|---|---|---|
| Myocardial dysfunction | 160 (10%) | 16 (11%) | 33 (14%) | 402 (37%) | 131 (21%) | 43 (13%) |
| Cardiac arrest before admission | 57 (4%) | 5 (3%) | 7 (3%) | 120 (11%) | 17 (3%) | 9 (3%) |
| Hyperglycemia (>11 mmol/l) | 164 (10%) | 22 (15%) | 39 (16%) | 299 (27%) | 82 (13%) | 73 (22%) |
| Hypoglycemia (<3 mmol/l) | 13 (1%) | 5 (3%) | 5 (2%) | 85 (8%) | 8 (1%) | 8 (2%) |
| Body temperature (°C) | 38.3 [37.7–39] | 38.9 [37.8–39.5] | 37.7 [37.2–38.2] | 38 [37.3–38.6] | 38.0 [37.4–38.7] | 38.8 [38–39.6] |
| New atrial fibrillation | 212 (13%) | 20 (13%) | 21 (9%) | 173 (16%) | 123 (20%) | 96 (28%) |
| Recurrent atrial fibrillation | 42 (3%) | 4 (3%) | 11 (5%) | 126 (12%) | 40 (6%) | 10 (3%) |
| Heart rate (beats/min) | 115 [100–130] | 115 [97–130] | 110 [100–124] | 114 [98–135] | 118 [100–138] | 129 [112–147] |
| Respiratory rate (breaths/min) | 26 [20–33] | 23 [20–30] | 27 [20–33] | 24 [20–31] | 20 [16–25] | 29 [24–35] |
| Sodium blood level (mmol/l) | 137 [133–141] | 136 [132–140] | 138 [135–141] | 137 [133–142] | 137 [133–141] | 136 [132–140] |
| Potassium blood level (mmol/l) | 3.9 [3.5–4.4] | 3.6 [3.3–4.2] | 4.2 [3.8–4.8] | 4.2 [3.5–5] | 4.2 [3.7–4.8] | 3.7 [3.2–4.2] |
| Bicarbonate blood level (mmol/l) | 23 [19–26] | 22 [18–25] | 27 [22–33] | 18.6 [14–23] | 19 [15–23] | 20 [16–24] |
| Hematocrit (%) | 35 [30–40] | 37 [31–41] | 40 [34–45] | 32 [28–36] | 31 [27–36] | 25 [22–28] |
| Prothrombin time (%) | 74 [62–85] | 74 [63–85] | 78 [59–92] | 56 [40–71] | 59 [47–70] | 59 [48–73] |
| Leukocytes (x10³/mm3), | 12.6 [8.4–17.8] | 12.5 [7.9–19.9] | 12.5 [9.2–17.2] | 14.8 [9.6–21.1] | 12.800 [7.9–18.9] | 0.8 [0.2–4.3] |
| Fluid replacement >50 ml/kg | 242 (15%) | 28 (19%) | 12 (5%) | 253 (23%) | 157 (25%) | 61 (18%) |

*Definition of abbreviations*: COPD = chronic obstructive pulmonary disease; HIV = human immunodeficiency virus; AIDS = acquired immune deficiency syndrome; IQR = interquartile range; **Cluster 1** = young patients without any comorbidities, admitted in ICU for community-acquired pneumonia; **Cluster 2** = young patients without any comorbidities, admitted in ICU for meningitis or encephalitis; **Cluster 3** = elderly patients with COPD, admitted in ICU for bronchial infection with few organ failures; **Cluster 4** = elderly patients, with several comorbidities and organ failures; **Cluster 5** = patients admitted after surgery with a nosocomial infection; **Cluster 6** = young patients with immunosuppressive disease or therapy, such as AIDS, chronic steroid therapy or hematological malignancy. Values in Numbers (%) or median [IQR].

## Outcome data

The survival curves shown in Fig 2 were significantly different between clusters (log-rank test: p <0.01). The risks of early, intermediate and late mortality, with and without adjustment, are shown in Fig 3. Results remained similar in the sub-groups limited to patients with septic shock (S7 Fig). The differences in length of ICU and hospital stay, number of ventilator-free

**Table 4. Organ failure at admission by cluster (performed in training set).**

| Variable | Cluster 1 n = 1,603 | Cluster 2 n = 149 | Cluster 3 n = 243 | Cluster 4 n = 1,094 | Cluster 5 n = 623 | Cluster 6 n = 338 |
|---|---|---|---|---|---|---|
| Vasopressor at admission | 714 (45%) | 58 (39%) | 68 (28%) | 799 (73%) | 470 (75%) | 204 (60%) |
| Glasgow Coma Score | 15 [8–15] | 9 [6–13] | 15 [13–15] | 13 [7–15] | 15 [13–15] | 15 [13–15] |
| Creatinine level (µmol/l) | 85 [64–120] | 86 [72–121] | 80 [62–114] | 165 [110–278] | 111.5 [76.5–185] | 111.5 [76.5–185] |
| Platelets count (x10³/mm³) | 223 [155–300] | 209 [125–282] | 234 [182–308] | 182 [110–275] | 214.5 [132–297] | 40 [24–80] |
| PaO2/FiO2 ratio (mmHg) | 217 [133–378] | 312 [214–493] | 240 [170–322] | 224 [136–358] | 267 [180–382] | 315 [168–497] |
| Bilirubin level (mmol/l) | 11 [7–19] | 14 [8–24] | 9 [6–13] | 16 [9–30] | 17 [10–30] | 20 [11–42] |
| Blood lactate level (mmol/l) | 1.7 [1.2–2.7] | 1.8 [1.3–3.2] | 1.5 [1.1–2.2] | 2.6 [1.6–4.9] | 2.2 [1.4–3.7] | 2.2 [1.5–4.4] |

*Definition of abbreviations*: COPD = chronic obstructive pulmonary disease; HIV = human immunodeficiency virus; AIDS = acquired immune deficiency syndrome; IQR = interquartile range; **Cluster 1** = young patients without any comorbidities, admitted in ICU for community-acquired pneumonia; **Cluster 2** = young patients without any comorbidities, admitted in ICU for meningitis or encephalitis; **Cluster 3** = elderly patients with COPD, admitted in ICU for bronchial infection with few organ failures; **Cluster 4** = elderly patients, with several comorbidities and organ failures; **Cluster 5** = patients admitted after surgery with a nosocomial infection; **Cluster 6** = young patients, with immunosuppressive disease or therapy, such as AIDS, chronic steroid therapy or hematological malignancy. Values in Numbers (%) or median [IQR].

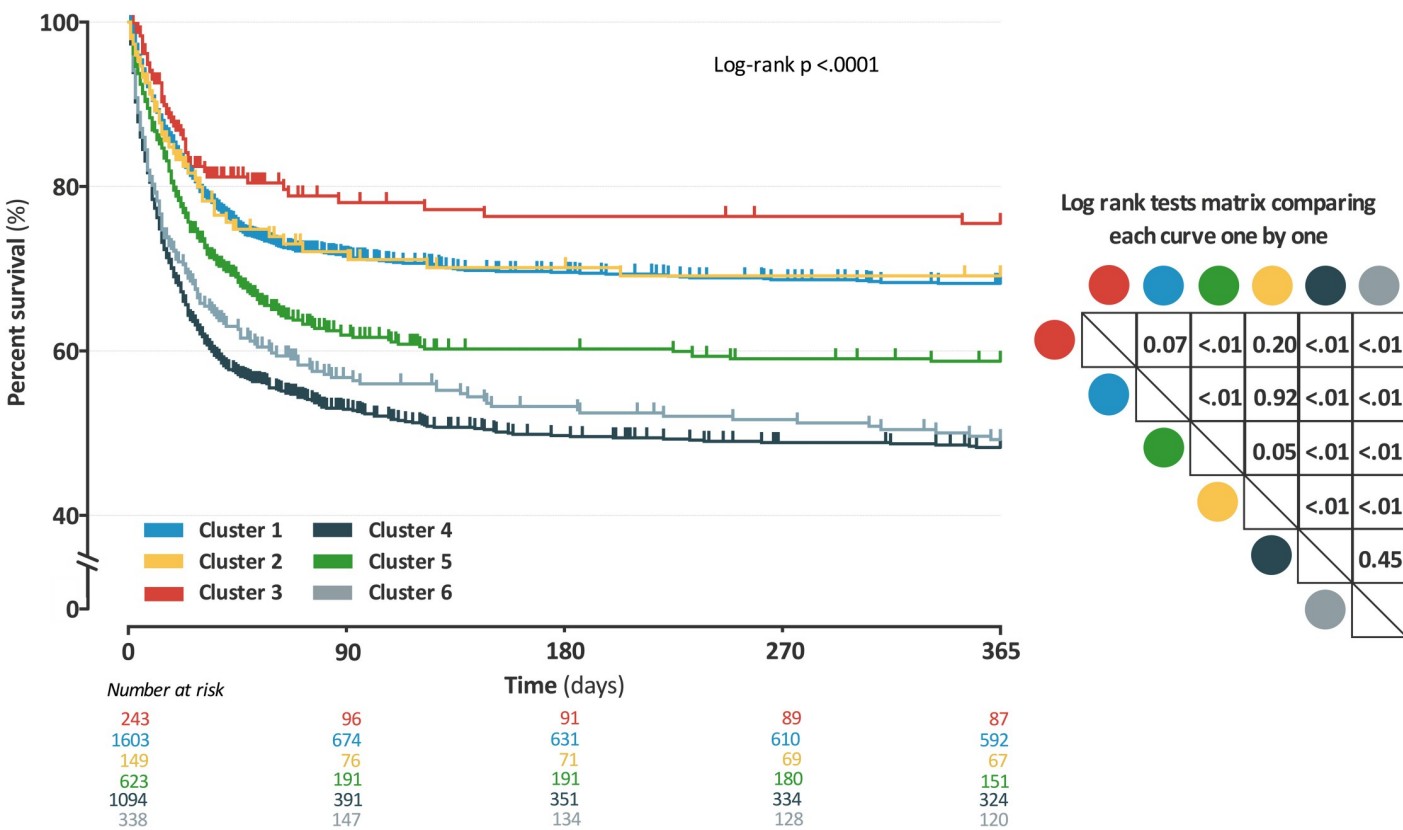

**Fig 2. Mortality estimated by Kaplan-Meier according to the cluster assignment with log rank tests (performed in training set).** *Definition of abbreviations*: Each curve was compared one by one using a log rank test; Results of these tests are presented in a double entry matrix; each cluster can be identified by its color; Analysis was performed including all patients in sepsis or septic shock. **Cluster 1** = young patients without any comorbidities, admitted in ICU for community-acquired pneumonia; **Cluster 2** = young patients without any comorbidities, admitted in ICU for meningitis or encephalitis; **Cluster 3** = elderly patients with COPD, admitted in ICU for bronchial infection with few organ failures; **Cluster 4** = elderly patients with several comorbidities and organ failures; **Cluster 5** = patients admitted after surgery with a nosocomial infection; **Cluster 6** = young patients with immunosuppressive disease or therapy, such as AIDS, chronic steroid therapy or hematological malignancy.

days at 28-day, duration of renal replacement therapy, number of catecholamine-free days at 28-day, or of organ system failure-free days at 28-day are described in S2 Table.

## Binary tree

Six discriminatory variables available on admission were identified by CART methods to assign new patients into a cluster (S8 Fig). The distribution of the patients amongst the clusters was similar. The accuracy of the binary tree in the training set is shown on S3 Table.

## Cluster validation

Six clusters were identified in the validation set (S9 Fig). The contributions of variables in the construction of the first four dimensions of the MCA are depicted on S10 Fig. S4 Table describe the distribution of demographic characteristics, various comorbidities, sources of infection, micro-organisms, clinical and biological data, and the organ failures at admission. The accuracy of the binary tree in the validation set is shown on S5 Table. The results of the clusters analysis after exclusion of the oldest data (admission before 2008) are provided in S6 Table.

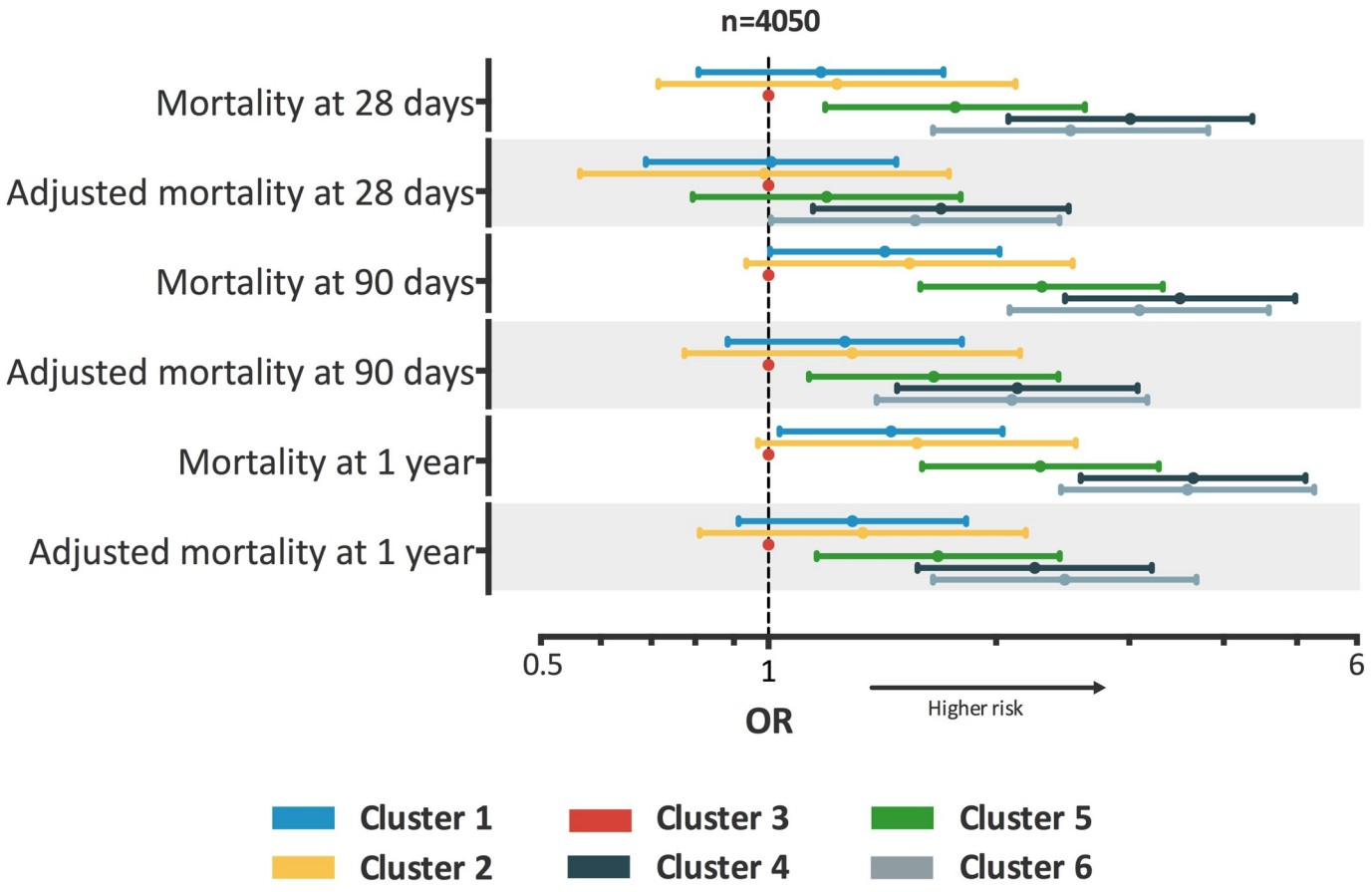

**Fig 3. Description of the clusters and their risks of early, intermediate and late mortality, with and without adjustment (performed in training set).** *Definition of abbreviations*: HR: Hazard ratio; The analysis was performed after exclusion of patients without septic shock. A Cox model was used to determine the hazard ratio; Data are reported as HR ± 95% confidence intervals, presented from lowest to highest;presented from lowest to highest; Cluster 3 was used as reference; Adjusted mortality were adjusted using SOFA score at admission and year of ICU admission. **Cluster 1** = young patients without any comorbidities, admitted in ICU for community-acquired pneumonia; **Cluster 2** = young patients without any comorbidities, admitted in ICU for meningitis or encephalitis; **Cluster 3** = elderly patients with COPD, admitted in ICU for bronchial infection with few organ failures; **Cluster 4** = elderly patients with several comorbidities and organ failures; **Cluster 5** = patients admitted after surgery with a nosocomial infection; **Cluster 6** = young patients with immunosuppressive disease or therapy such as AIDS, chronic steroid therapy or hematological malignancy.

## Discussion

Our work is an original clinical study based on a large sample that tried to reduce the heterogeneity of septic population. Using the new definition of sepsis 3.0 [4], we performed a HC based on clinical and biological data commonly available at ICU admission. We were able to discriminate 6 rather homogeneous clusters of patients with sepsis and septic shock. Three clusters were characterized by underlying disorders, while two clusters were characterized by the source of infection. Baseline risk of death at day-28, day-90, and one year were significantly different across clusters, independently of organ failures at admission. After having developed a binary classification tree, we were able to identify similar results from the validation subset. The ability to affect a patient with sepsis to a homogeneous cluster should enable to achieve a personalized ICU medical care strategy, and to test potentially appropriate new therapies by performing more efficiently targeted clinical trials. The simplest way to do so is to use classification tree.

The low influence of organ dysfunction to characterize clusters was the main result that was comparable with the study of Seymour *et al* [18]. Two clusters share identical characteristics, one that included young patients with low comorbidity, and one that included elderly patients with high Charlson scores. Some similarities can be found between their δ clusters and our cluster 5. Patients had more liver dysfunction, and more often a septic shock and intra-abdominal sepsis. It was the most frequent site of infection of this cluster. Interestingly, we identified a well-defined cluster including immunocompromised patients, that mainly comprised hematologic patients. These patients were excluded of the SENECA cohorts, GenIMS cohort, and of the ProCESS trial, PROWESS trial or ACCESS trial.

We identified a small cluster that included patients with COPD exacerbation. These patients met to the definition of sepsis 3.0, yet mechanisms of organ failure were probably very different from other cluster. The inclusion of these patients in sepsis studies should be discussed.

Some limitations must be acknowledged for this study. First, these phenotypes will reduce heterogeneity prior to randomization; however, because they will not necessarily create better trials, further studies are necessary to better explain this clustering [31]. Second, the training set and the validation set come from the same database. A validation with an external database would have been more robust. Third, the time of patient's admission on our study ranged from 1997 to 2015 with heterogeneous periods of inclusion and number of patients included between each center (S7 Table). The improvement of the prognosis related to the improvements in sepsis management is no longer debated. This difference in risks of mortality was taken into account by an adjustment on the year of ICU admission and a sensitivity analysis was performed without the oldest data. Finally, unsupervised analysis is not unbiased because only the data available within the database can be explored, and only the data that are known or thought to be important are entered [25]. Some data would have been necessary, in particular data about clinical management or specific therapy [26]. Also, there is a lack of information about cytokines, ethnicity, genetic polymorphism, precise dating of the infection onset, and biomarkers like C-reactive protein test. Several studies have focused on identification of sepsis molecular phenotypes based on gene expression data [16,27,28]. Between two to four clusters were identified. Genotype or endotype approach has the potential to substantially improve our understanding of the key biological pathways involved in human diseases and to suggest new targets for treatment or prevention. However, studies using these approaches often failed to replicate positive findings, especially when investigating associations with sepsis outcomes [29,30]. The possible explanations include low statistical power, heterogeneous patient populations, and imprecise definition of phenotypes [31]. As suggested in a review by Clark *et al*, sepsis defines a syndrome rather than a specific disease. This may lead to a marked heterogeneity in patient populations, which may explain some of the variations in the findings. Also, he suggested to use more precise phenotypes, for example, meningococcal sepsis or fecal peritonitis. According to Rautanen *et al*, promising results might be generated by focusing on more homogenous subgroups such as sepsis patients with pneumonia [32]. The understanding of the immune system and its interaction with pathogens must take into account the high dimensionality of the data, future studies are needed to aggregate different data in order to combine clinical data, host genomics, transcriptomic responses and cytokines, and using data science approaches accounting for longitudinal data.

When the hypothesis is made that the treatment effect is similar in all patients and evolves linearly with the severity of illness, the expected effect of the new treatment in a randomized trial measured by the reduction of the risk of death is dependent of the baseline risk of death and its distribution in the sample. This was described and illustrated by Kent [33]. In their simulation study of therapeutic trials on ARDS and sepsis, Iwashyna *et al* showed that the

variation in the baseline risk of death was the main determinant of the heterogeneity of the treatment response [34]. Thus, the impact of the same treatment applied to populations with a different baseline risk of death spanned from an increase in the risk of death in a low risk population to a decrease in the risk of death in patients whom baseline risk of death was high. However, patients with sepsis are intrinsically heterogeneous, not only in their baseline risk of death [35,36], but also in their risk of adverse outcome [37]. Identifying treatable traits and setting an accurate diagnosis are the major challenges in sepsis [37]. Improving prognostic estimation, and performing comparisons and benchmarking of processes and outcomes between different ICUs are also necessary. Although the current definition of sepsis groups was designed to reduce heterogeneity of the patients [4], an important variability can still be observed for many patients' characteristics such as the sources of infection, causative pathogen (s), age, lifestyle, comorbidities, or genetic profile.

## Conclusion

Because the prognostication and the identification of target patients remain difficult in sepsis, new approaches are necessary. In patients who met the new sepsis 3.0 definition, six clusters, clearly different in their clinical and biological presentation, were identified by using hierarchical clustering. These clusters also differed in mortality and severity of illness. Considering these clusters may reduce the uncontrolled differences in patients' prognosis and improve the power of studies. Future works including big data analysis, clinical and genomic data and several biomarkers may contribute to better defining homogeneous subsets of sepsis patients.

## Supporting information

**S1 Fig. Flow chart.**
(DOCX)

**S2 Fig. Dendrogram of ascending hierarchical clustering analysis used in the training set.**
Dendrogram obtained after application of hierarchical clustering analysis by accounting for the 51 dimensions of the multiple correspondence analysis. The vertical axis of the dendrogram represents the distance between clusters. The horizontal vertical axis represents the patients and clusters. Each junction between two clusters is represented on the graph by the split of a vertical line into two vertical lines. The vertical position of the split, shown by the short horizontal bar, gives the distance between the two clusters. The red line shows the cut level that determines the number of clusters. The indices used to determine this cut level, Semi partial R-Squared, the Squared-R, the Pseudo F statistic and the Pseudo t2 statistic, are presented in S3 Fig.
(DOCX)

**S3 Fig. Reprentation of indices used to determine the dendrogram cut level. A**: The Semi partial R-Squared given the decrease in the proportion of variance accounted of resulting from joining the two clusters. **B**: The Squared-R is the proportion of variance accounted for by the cluster. **C**: The Pseudo F statistic measuring the separation among all clusters at the current level. **D**: The Pseudo t2 statistic measuring the separation between the two clusters most recently joined. There is no consensus in the literature on the final choice of the number of clusters regardless of the clustering method. According to the different criteria (Semi partial R-Squared = 4 to 6 clusters, R squared = upper 10, Pseudo F statistic = 4 to 7 clusters, Pseudo $t^2$ statistic = 2, 4 or 6 clusters), the number of six clusters is the best trade-off between goodness of fit criteria and clinical interpretation.
(DOCX)

**S4 Fig. Contribution of each variable to the first four dimensions of the MCA.** *Definition of abbreviations*: MCA = multiple correspondence analysis; COPD = chronic obstructive pulmonary disease; ICU = intensive care unit; SOFA = Sequential Organ Failure Assessment score; The contribution is expressed as a percentage. The representation is limited to the first 20 variables that contribute most to the dimension of the 63 used. The variables represented are classified in descending order of contribution.
(DOCX)

**S5 Fig. Point cloud of patients representing the 6 clusters in the first four dimensions of the MCA.** Definition of abbreviations: MCA: Multiple correspondence analysis; HC: Hierarchical clustering; Each patient was represented using his individual coordinates in these dimensions. Patient's assignment to a cluster was obtained after HC application. **A:** Axes correspond to the first and second dimension of MCA. Each patient was represented using his individual coordinates in these dimensions. **B:** Axes correspond to the third and fourth dimension of MCA.
(DOCX)

**S6 Fig. Probability for variables for belonging to one cluster assessed using univariable logistic regression. A**: Figure of cluster 1; B: Figure of cluster 2; C: Figure of cluster 3; D: Figure of cluster 4; E: Figure of cluster 5; F: Figure of cluster 6; Definition of abbreviations: COPD = chronic obstructive pulmonary disease; HIV = human immunodeficiency virus; AIDS = acquired immune deficiency syndrome; Leucopenia was defined by a leukocyte count below 500/mm3; Hyperlactatemia was defined by a blood lactate concentration greater than 2 mmol/l; Median of SOFA score by organ was used to define organ dysfunction in binary variable. Associations with leucopenia and bronchial could not be calculated due to insufficient numbers in this cluster.
(DOCX)

**S7 Fig. Risks of mortality at 28 days, 90 days and one-year according to the cluster assignment in patients with septic shock in the training set.** *Definition of abbreviations*: OR: Odds ratio; The analysis was performed after exclusion of patients without septic shock. A logistic regression was used to determine the odds ratio; Data are reported as odds ratios ± 95% confidence intervals, presented from lowest to highest; Cluster 3 was used as reference class; Adjusted mortality were adjusted using SOFA score at admission and year of ICU admission. **Cluster 1** = young patients without any comorbidities, admitted in ICU for community-acquired pneumonia; **Cluster 2** = young patients without any comorbidities, admitted in ICU for meningitis or encephalitis; **Cluster3** = elderly patients with COPD, admitted in ICU for bronchial infection with few organ failures; **Cluster 4** = elderly patients with several comorbidities and organ failures; **Cluster 5** = patients admitted after surgery with a nosocomial infection; **Cluster 6** = young patients with immunosuppressive disease or therapy, such as AIDS, chronic steroid therapy or hematological malignancy.
(DOCX)

**S8 Fig. Binary tree using variables available at the ICU admission.** *Definition of abbreviations*: ICU = intensive care unit; The binary tree was built in the training set using Breiman methods [22] with Rpart package version 4.1–10 [23], R version 3.1.0. The structure is similar to a real tree, from the bottom up: There is a root, where the first split happens. After each split, two new nodes are created. Each node contains only a subset of the patients with a minimum size of 20. The partitions of the data, which are not split any more, are called terminal nodes or leafs. The second stage of the procedure consists to prune the tree using cross-validation. Pruning means to shorten the tree, which makes trees more compact and avoids over

fitting to the training data. Each split is examined, if it brings a reliable improvement. The six variables used by the binary tree are: Lung infection, Surgical admission, Hematological malignancy, Bronchial infection, Chronic heart failure and Meningeal infection. The accuracy of the binary tree evaluated in the training dataset was available on S3 Table. **Cluster 1** = young patients without any comorbidities, admitted in ICU for community-acquired pneumonia; **Cluster 2** = young patients without any comorbidities, admitted in ICU for meningitis or encephalitis; **Cluster3** = elderly patients with COPD, admitted in ICU for bronchial infection with few organ failures; **Cluster 4** = elderly patients with several comorbidities and organ failures; **Cluster 5** = patients admitted after surgery with a nosocomial infection; **Cluster 6** = young patients with immunosuppressive disease or therapy, such as AIDS, chronic steroid therapy or hematological malignancy.
(DOCX)

**S9 Fig. Representation of indices used to determine the dendrogram cut level (performed in validation set).** A: The Semi partial R-Squared given the decrease in the proportion of variance accounted of resulting from joining the two clusters. B: The Squared-R is the proportion of variance accounted for by the cluster. C: The Pseudo F statistic measuring the separation among all clusters at the current level. D: The Pseudo t2 statistic measuring the separation between the two clusters most recently joined. There is no consensus in the literature on the final choice of the number of clusters regardless of the clustering method. According to the different criteria (Semi partial R-Squared = 3 to 6 clusters, R squared = upper 10, Pseudo F statistic = 2 to 7 clusters, Pseudo t2 statistic = 3 or 6 clusters), the number of six clusters is the best trade-off between goodness of fit criteria and clinical interpretation.
(DOCX)

**S10 Fig. Contribution of each variable to the first four dimensions of the MCA (performed in validation set).** Definition of abbreviations: MCA = multiple correspondence analysis; COPD = chronic obstructive pulmonary disease; ICU = intensive care unit; SOFA = Sequential Organ Failure Assessment score; The contribution is expressed as a percentage. The representation is limited to the first 20 variables that contribute most to the dimension of the 63 used. The variables represented are classified in descending order of contribution.
(DOCX)

**S1 Table. The 63 variables included in the cluster analysis, distribution and missing values.** Definition of abbreviations: COPD = chronic obstructive pulmonary disease; HIV = human immunodeficiency virus; AIDS = acquired immune deficiency syndrome; NF GNB: Non-fermentative Gram negative bacilli; MDRO: Multi-drug resistance organisms (refer to vancomycin-resistant enterococci, methicillin-resistant Staphylococcus aureus, extended-spectrum β-lactamase-producing Enterobacteriaceae, AmpC-producing Enterobacteriaceae, Pseudomonas aeruginosa resistant to more than two antimicrobial families, Stenotrophomonas maltophilia); Values in Numbers (%) or median [IQR].
(DOCX)

**S2 Table. Outcomes of patients (performed in training set).** Definition of abbreviations: ICU = intensive care unit; IQR = interquartile range; Free days were censored at 28 days; A ventilator-free day refers to a day without invasive or non-invasive mechanical ventilation or death; A catecholamine-free day refers to a day without to vasoactive or inotropic agent or death; An organ system failure-free day refers to a day without SOFA score upper zero or death; Cluster 1 = young patients without any comorbidities, admitted in ICU for community-acquired pneumonia; Cluster 2 = young patients without any comorbidities, admitted in ICU

for meningitis or encephalitis; Cluster 3 = elderly patients with COPD admitted in ICU for bronchial infection with few organ failures; Cluster 4 = elderly patients, with several comorbidities and organ failures; Cluster 5 = patients admitted after surgery with a nosocomial infection; Cluster 6 = young patients with immunosuppressive disease or therapy, such as AIDS, chronic steroid therapy or hematological malignancy. Values in Numbers (%) or median [IQR]. P- values were obtained by Analysis of variance or Chi-2 test.
(DOCX)

**S3 Table. Accuracy of the binary tree (performed in training set).** Definition of abbreviations: Truly assigned = Number of patients correctly assigned as belonging to the cluster according to the total number of patients in the cluster; Falsely assigned = Number of patients incorrectly labeled as belonging to the cluster according to the total number of patients out of the cluster; Se = Sensitivity; Sp = Specificity; AUC = Area under the Receiver Operating Characteristic curve; IC95% = 95% confidence intervals; Cluster 1 = young patients, without any comorbidities, admitted in ICU for community-acquired pneumonia; Cluster 2 = young patients, without any comorbidities, admitted in ICU for meningitis or encephalitis; Cluster 3 = elderly patients with COPD, admitted in ICU for bronchial infection with few organ failures; Cluster 4 = elderly patients, with several comorbidities and organ failures; Cluster 5 = patients admitted after surgery with a nosocomial infection; Cluster 6 = Young patients, with immunosuppressive disease or therapy, such as AIDS, chronic steroid therapy or hematological malignancy.
(DOCX)

**S4 Table. Characteristics of patients set after assignment in cluster by the cluster analysis (performed in validation set).** Definition of abbreviations: COPD = chronic obstructive pulmonary disease; HIV = human immunodeficiency virus; AIDS = acquired immune deficiency syndrome; NF GNB: Non-fermentative Gram negative bacilli; MDRO: Multi-drug resistance organisms (refer to vancomycin-resistant enterococci, methicillin-resistant Staphylococcus aureus, extended-spectrum β-lactamase-producing Enterobacteriaceae, AmpC-producing Enterobacteriaceae, Pseudomonas aeruginosa resistant to more than two antimicrobial families, Stenotrophomonas maltophilia); Values in Numbers (%) or median [IQR].
(DOCX)

**S5 Table. Accuracy of the binary tree (performed in validation set).** Definition of abbreviations: Truly assigned = Number of patients correctly assigned as belonging to the cluster according to the total number of patients in the cluster; Falsely assigned = Number of patients incorrectly labeled as belonging to the cluster according to the total number of patients out of the cluster; Se = Sensitivity; Sp = Specificity; AUC = Area under the Receiver Operating Characteristic curve; IC95% = 95% confidence intervals; Cluster 1 = young patients, without any comorbidities, admitted in ICU for community-acquired pneumonia; Cluster 2 = young patients, without any comorbidities, admitted in ICU for meningitis or encephalitis; Cluster 3 = elderly patients with COPD, admitted in ICU for bronchial infection with few organ failures; Cluster 4 = elderly patients, with several comorbidities and organ failures; Cluster 5 = patients admitted after surgery with a nosocomial infection; Cluster 6 = Young patients, with immunosuppressive disease or therapy, such as AIDS, chronic steroid therapy or hematological malignancy.
(DOCX)

**S6 Table. Characteristics of patients after excluding the oldest data (i.e., admission before 2008) (performed in training set).** Definition of abbreviations: COPD = chronic obstructive pulmonary disease; HIV = human immunodeficiency virus; AIDS = acquired immune

deficiency syndrome; NF GNB: Non-fermentative Gram negative bacilli; MDRO: Multi-drug resistance organisms (refer to vancomycin-resistant enterococci, methicillin-resistant Staphylococcus aureus, extended-spectrum β-lactamase-producing Enterobacteriaceae, AmpC-producing Enterobacteriaceae, Pseudomonas aeruginosa resistant to more than two antimicrobial families, Stenotrophomonas maltophilia); Values in Numbers (%) or median [IQR].
(DOCX)

**S7 Table. Number of patients included and median of admission in each center.**
(DOCX)

**S1 File. Dataset.**
(XLS)

## Acknowledgments

The authors thank Celine Feger (MD, EMIBiotech) for her editorial support.

## ^Members of the OUTCOMEREA Study Group

**Correspondant:** Jean-François Timsit (Medical and Infectious Diseases ICU, Bichat-Claude Bernard Hospital, Paris, France; UMR 1137 Inserm–Paris Diderot university IAME, F75018, Paris)–jean.francois.timsit@aphp.fr

**Scientific Committee:** Jean-François Timsit (Medical and Infectious Diseases ICU, Bichat-Claude Bernard Hospital, Paris, France; UMR 1137 Inserm–Paris Diderot university IAME, F75018, Paris); Elie Azoulay (Medical ICU, Saint Louis Hospital, Paris, France); Maïté Garrouste-Orgeas (ICU, Saint-Joseph Hospital, Paris, France); Jean-Ralph Zahar (Infection Control Unit, Angers Hospital, Angers, France); Christophe Adrie (Physiology, Cochin Hospital, Paris, France); Michael Darmon (Medical ICU, Saint Etienne University Hospital, St Etienne, France); and Christophe Clec'h (ICU, Avicenne Hospital, Bobigny, and UMR 1137 Inserm–Paris Diderot university IAME, F75018, Paris, France).

**Biostatistical and Information System Expertise:** Jean-Francois Timsit (Medical and Infectious Diseases ICU, Bichat-Claude Bernard Hospital, Paris, France; UMR 1137 Inserm–Paris Diderot university IAME, F75018, Paris); Corinne Alberti (Medical Computer Sciences and Biostatistics Department, Robert Debré Hospital, Paris, France); Adrien Français (Integrated Research Center U823, Grenoble, France); Aurélien Vesin (OUTCOMEREA organization and Integrated Research Center U823, Grenoble, France); Stephane Ruckly (OUTCOMEREA organization and Inserm UMR 1137 IAME, F75018, Paris); Sébastien Bailly (Grenoble university hospital Inserm UMR 1137 IAME, F75018, Paris) and Christophe Clec'h (ICU, Avicenne Hospital, Bobigny, and Inserm UMR 1137 IAME, F75018, Paris, France); Frederik Lecorre (Supelec, France); Didier Nakache (Conservatoire National des Arts et Métiers, Paris, France); and Aurélien Vannieuwenhuyze (Tourcoing, France).

**Investigators of the OUTCOMEREA Database:** Dr Romain HERNU

Christophe Adrie (ICU, CH Melun, and Physiology, Cochin Hospital, Paris, France); Carole Agasse (medical ICU, university hospital Nantes, France); Bernard Allaouchiche (ICU, Pierre benite Hospital, Lyon, France); Olivier Andremont (ICU, Bichat Hospital, Paris, France); Pascal Andreu (CHU Dijon, Dijon, France); Laurent Argaud (Medical ICU, Hospices Civils de Lyon, Lyon, France); Claire Ara-Somohano (Medical ICU, University Hospital, Grenoble, France); Elie Azoulay (Medical ICU, Saint Louis Hospital, Paris, France); Francois Barbier (medical-surgical ICU, Orleans, France), Déborah Boyer (ICU, CHU Rouen, France), Jean-Pierre Bedos (ICU, Versailles Hospital, Versailles, France); Thomas Baudry (Medial ICU, Edouard Heriot hospital, Lyon France), Jérome Bedel (ICU, Versailles Hospital, Versailles,

France), Julien Bohé (ICU, Hôpital Pierre Benite, Lyon France), Lila Bouadma (ICU, Bichat Hospital, Paris, France); Jeremy Bourenne (Réanimation des urgences, Timone-2; APHM, Marseille, France); Noel Brule (medical ICU, university hospital Nantes, France); Cédric Brétonnière (medical ICU, university hospital Nantes, France); Christine Cheval (ICU, Hyeres Hospital, Hyeres, France); Julien Carvelli (Réanimation des urgences, Timone-2; APHM, Marseille, France);Christophe Clec'h (ICU, Avicenne Hospital, Bobigny, France); Elisabeth Coupez (ICU, G Montpied Hospital, Clermont-Ferrand, France); Martin Cour Medial ICU, Edouard Heriot hospital, Lyon France), Michael Darmon (ICU, Saint Etienne Hospital, Saint Etienne, France); Etienne de Montmollin (ICU, Delafontaine Hospital, Saint Denis), Loa Dopeux (ICU, G Montpied Hospital, Clermont-Ferrand, France); Anne-Sylvie Dumenil (Antoine Béclère Hospital, Clamart, France); Claire Dupuis (Bichat hospital and UMR 1137 Inserm–Paris Diderot university IAME, F75018, Paris, France), Jean-Marc Forel (AP HM, Medical ICU, Hôpital Nord Marseille), Marc Gainnier (Réanimation des urgences, Timone-2; APHM, Marseille, France), Charlotte Garret (medical ICU, university hospital Nantes, France); Dany Goldgran-Tonedano (CH le Raincy-Montfermeil; France); Steven Grangé (ICU, CHU Rouen, France), Antoine Gros (ICU, Versailles Hospital, Versailles, France), Hédia Hammed (CH le Raincy-Montfermeil); Akim Haouache (Surgical ICU, H Mondor Hospital, Creteil, France); Romain Hernu (Medical ICU, Hospices Civils de Lyon, Lyon, France); Tarik Hissem (ICU, Eaubonne, France), Vivien Hon Tua Ha (ICU, CH Meaux, France); Sébastien Jochmans (ICU, CH Melun); Jean-Baptiste Joffredo (ICU, G Montpied Hospital, Clermont-Ferrand, France); Hatem Kallel (ICU, Cayenne General Hospital, Cayenne, France); Guillaume Lacave (ICU, Versailles Hospital, Versailles, France), Alexandre Lautrette (ICU, G Montpied Hospital, Clermont-Ferrand, France); Virgine Lemiale (Medical ICU, Saint Louis Hospital, Paris, France); Mathilde Lermuzeaux (ICU, Bichat Hospital, Paris, France), Guillaume Marcotte (Surgical ICU, Hospices Civils de Lyon, Lyon, France); Jordane Lebut (ICU, Bichat Hospital, Paris, France); Maxime Lugosi (Medical ICU, University Hospital Grenoble, Grenoble, France); Eric Magalhaes (ICU, Bichat Hospital, Paris, France), Sibylle Merceron (ICU, Versailles Hospital, Versailles, France), Bruno Mourvillier (ICU, Bichat Hospital, Paris, France); Benoît Misset (ICU, Saint-Joseph Hospital, Paris, France and Medical ICU CHU Rouen, France); Bruno Mourvillier (ICU, Bichat Hospital, Paris, France); Mathild Neuville (ICU, Bichat Hospital, Paris, France), Laurent Nicolet (medical ICU, university hospital Nantes, France); Johanna Oziel (Medico-surgical ICU, hôpital Avicenne APHP, Bobigny, France), Laurent Papazian (Hopital Nord, Marseille, France), Benjamin Planquette (pulmonology ICU, George Pompidou hospital Hospital, Paris, France); Jean-Pierre Quenot (CHU Dijon, Dijon, France); Aguila Radjou (ICU, Bichat Hospital, Paris, France), Marie Simon (Medial ICU, Edouard Heriot hospital, Lyon France), Romain Sonneville (ICU, Bichat Hospital, Paris, France), Jean Reignier (medical ICU, university hospital Nantes, France); Bertrand Souweine (ICU, G Montpied Hospital, Clermont-Ferrand, France); Carole Schwebel (ICU, A Michallon Hospital, Grenoble, France); Shidasp Siami (ICU, Eaubonne, France); Roland Smonig (ICU, Bichat Hospital, Paris, France); Gilles Troché (ICU, Antoine Béclère Hospital, Clamart, France); Marie Thuong (ICU, Delafontaine Hospital, Saint Denis, France); Guillaume Thierry (ICU, Saint-Louis Hospital, Paris, France); Guillaume Van Der Meersch–Medical Surgical ICU, university hospital Avicenne), Marion Venot (Medical ICU, Saint Louis Hospital, Paris, France); Sondes Yaacoubi (CH le Raincy-Montfermeil); Olivier Zambon (medical ICU, university hospital Nantes, France);

**Study Monitors**: Julien Fournier, Caroline Tournegros, Stéphanie Bagur, Mireille Adda, Vanessa Vindrieux, Sylvie de la Salle, Pauline Enguerrand, Loic Ferrand, Vincent Gobert, Stéphane Guessens, Helene Merle, Nadira Kaddour, Boris Berthe, Samir Bekkhouche, Kaouttar Mellouk, Mélaine Lebrazic, Carole Ouisse, Diane Maugars, Christelle Aparicio, Igor Theodose,

Manal Nouacer, Veronique Deiler, Myriam Moussa, Atika Mouaci, Nassima Viguier and Sophie Letrou.

## Author Contributions

**Conceptualization:** Grégory Papin, Sébastien Bailly, Stéphane Ruckly, Jean-François Timsit.

**Data curation:** Grégory Papin, Claire Dupuis, Marc Gainnier, Laurent Argaud, Elie Azoulay, Christophe Adrie, Bertrand Souweine, Dany Goldgran-Toledano, Guillaume Marcotte, Antoine Gros, Jean Reignier, Bruno Mourvillier, Jean-Marie Forel, Romain Sonneville, Anne-Sylvie Dumenil, Michael Darmon, Maité Garrouste-Orgeas, Carole Schwebel, Jean-François Timsit.

**Formal analysis:** Grégory Papin, Sébastien Bailly, Stéphane Ruckly, Jean-François Timsit.

**Investigation:** Grégory Papin, Sébastien Bailly, Jean-François Timsit.

**Methodology:** Grégory Papin, Sébastien Bailly, Claire Dupuis, Stéphane Ruckly, Jean-François Timsit.

**Project administration:** Grégory Papin, Sébastien Bailly, Stéphane Ruckly, Jean-François Timsit.

**Resources:** Grégory Papin, Sébastien Bailly, Stéphane Ruckly, Jean-François Timsit.

**Supervision:** Sébastien Bailly, Claire Dupuis, Stéphane Ruckly, Jean-François Timsit.

**Validation:** Sébastien Bailly, Claire Dupuis, Stéphane Ruckly, Jean-François Timsit.

**Visualization:** Sébastien Bailly, Stéphane Ruckly, Jean-François Timsit.

**Writing – original draft:** Grégory Papin, Sébastien Bailly, Jean-François Timsit.

**Writing – review & editing:** Grégory Papin, Sébastien Bailly, Claire Dupuis, Stéphane Ruckly, Marc Gainnier, Laurent Argaud, Elie Azoulay, Christophe Adrie, Bertrand Souweine, Dany Goldgran-Toledano, Guillaume Marcotte, Antoine Gros, Jean Reignier, Bruno Mourvillier, Jean-Marie Forel, Romain Sonneville, Anne-Sylvie Dumenil, Michael Darmon, Maité Garrouste-Orgeas, Carole Schwebel, Jean-François Timsit.

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
