## [Decision Letter · Decision Letter 0]

19 Jan 2021

PONE-D-20-35126

Clinical and biological clusters of sepsis patients using hierarchical clustering.

PLOS ONE

Dear Dr. Papin,

Thank you for submitting your manuscript to PLOS ONE. After careful consideration, we feel that it has merit but does not fully meet PLOS ONE’s publication criteria as it currently stands. Therefore, we invite you to submit a revised version of the manuscript that addresses the points raised during the review process.

 Please submit your revised manuscript by Feb 18th. If you will need more time than this to complete your revisions, please reply to this message or contact the journal office at plosone@plos.org. Please include the following items when submitting your revised manuscript:

We look forward to receiving your revised manuscript.

Kind regards,

Martina Crivellari

Academic Editor

PLOS ONE

Journal Requirements:

2. Thank you for including your ethics statement: "According to the French law, the study was approved by institutional review board and hospital ethics committee, which waived the need for informed consent of patients included in the database."   

3. In the ethics statement in the manuscript and in the online submission form, please provide additional information about the patient records/samples used in your retrospective study, including: a) whether all data were fully anonymized before you accessed them; b) the date range (month and year) during which patients' medical records/samples were accessed; c) the date range (month and year) during which patients whose medical records/samples were selected for this study sought treatment.

4. One of the noted authors is a group or consortium [OUTCOMEREA network]. In addition to naming the author group, please list the individual authors and affiliations within this group in the acknowledgments section of your manuscript. Please also indicate clearly a lead author for this group along with a contact email address.

Additional Editor Comments:

In general, the paper is of interest, is presented in standard English, but there are many concerns  that have to be addressed. The paper is complex, there are data not always clear. The statistical analysis has not been performed appropriately and rigorously. The authors conclude that "[c]onsidering these clusters may reduce the uncontrolled differences in patients’ prognosis and improve the power of studies." It is not clear how the proposed clustering would help in clinical practice, authors should please clarify this point in the manuscript. There are no conflicts between the reviews.In my personal opinion even if this is an interesting paper, there are major questions to answer, and I was in doubt with the really possibility to make it achievable. I'm asking you a complete major revision with a specific attention to the statistical analysis. 

Reviewers' comments:

Reviewer's Responses to Questions

**Comments to the Author**

1. Is the manuscript technically sound, and do the data support the conclusions?

Reviewer #1: Partly

Reviewer #2: Partly

2. Has the statistical analysis been performed appropriately and rigorously? 

Reviewer #1: No

Reviewer #2: Yes

3. Have the authors made all data underlying the findings in their manuscript fully available?

Reviewer #1: Yes

Reviewer #2: No

4. Is the manuscript presented in an intelligible fashion and written in standard English?

Reviewer #1: Yes

Reviewer #2: Yes

5. Review Comments to the Author

Reviewer #1: In "Clinical and biological clusters of sepsis patients using hierarchical clustering." Papin et al. report on an analysis based on data from 6,046 sepsis/septic shock patients from the French national prospective multicenter ICU cohort OUTCOMEREA. Essentially, the authors performed a hierarchical clustering (HC) on a random sample of 4,050 patients and derived six clusters which differed in early or late mortality (even after adjustment on severity of organ dysfunctions (SOFA) and year of ICU admission). The authors conclude that "[c]onsidering these clusters may reduce the uncontrolled differences in patients’ prognosis and improve the power of studies."

Addressing the heterogeneity of sepsis patients to contribute e.g. to more targeted randomized controlled trials, is very important. The authors use retrospective data from a registry covering the period between 1997-2015. However, even after extensive reviewing, several major questions remain that have to be addressed. First of all the paper lacks clarity with regard to the steps that have been conducted. In this regard figure S1 and S2 should be put into the main text. Afterwards, it has to be clear in all figures and tables if the training or the validation data set is displayed. How many variables (overall) entered the HC – this number is important and is missing? If the HC is similarly conducted in the validation data set (independent of the training set) what cluster structure is derived and how similar is it to the six clusters? The focus should generally be on the validation data set! As this is the more critical step and of course generalizability to other ICU data sets is the ultimate step. Finally, and related to this point, the authors should extend the comparison to Seymour et al. (2019) and more systematically describe if and how their French results differ from those of the US electronic medical record analysis.

Other major points:

Page 8 (line 159-): What statistics are used for decisions and it is not clear why and how k-means clustering is included.

Table 1: Can be put to the Supplement – why did you categorize quantitative variables (which is not recommended!)

Page 10 (line 176-): These are time to event analyses (that can address censoring); thus, I hope that the mortality estimates are derived from the Kaplan Meier estimator (on page 11; line 206-); similarly I assume that Cox regression models were used for the analyses with adjustment?

Page 10 (line 187): Accuracy in the training set is of very little use – report this information in the validation set, please.

Page 10 (line 188-): Especially the red new text is not clear; in particular what is meant by “new”?

Page 11 (line 210-): What is meant by “The clusters supported 12% of the total variability.”?

Page 20 (line 333-): Is also not clear (I refer to the other points above).

Page 21 (line 358-): This recommendation should be attenuated and written more carefully.

Page 21 (line 364-): Please shorten the limitations part and structure the arguments better

Page 22 (line 391-): The argument regarding “selection bias” is not clear.

Page 23 (line 403-): “Second, …” where is first?

Minor points:

Page 5: The authors should mention the period of data collection already; moreover did the 20 ICUs contribute homogeneously along these 18 years? Please provide information on the ICUs their focus and temporal contribution

Page 7 (line 141-): Please provide a little more information on the reported coefficients

Page 7 (line 146-): Please provide more information on the missing value handling including the imputation and most importantly the impact that the imputation has on the conclusions

Page 23 (line 414): better use prognosis instead of “prognostication” throughout

Reviewer #2: Dear editor,

thank you very much for inviting me to review this paper.

The paper is a complex manuscript about the necessity to divide sepsis patients in cluster based on disease characteristics. Authors individuated six different clusters of patients analyzing data from a multicenter prospective database.

Even though I lack the expertise to judge the details of the methodology, the chosen approach to explain data is not clear and the deep use of figures and tables makes all the results incomprehensible. I would suggest explaining the results, figures, and tables in text to allow a better understanding.

Indeed, it is unclear what kind of information this study would add to daily clinical practice.

It is clearly evident how sepsis and septic shock are conditions deeply related to clinical characteristics of the host, microbes involved and concomitant diseases, but it is not clear how the proposed clustering would help in clinical practice, I would suggest to better clarify this point in paper.

Language is correct and all the bias and limits of the study are well indicated in manuscript.

6. PLOS authors have the option to publish the peer review history of their article (what does this mean?). If published, this will include your full peer review and any attached files.

Reviewer #1: **Yes: **André Scherag

Reviewer #2: **Yes: **Gaetano Lombardi

---

## [Author Response · Author response to Decision Letter 0]

30 Apr 2021

Editor Comments:

In general, the paper is of interest, is presented in standard English, but there are many concerns that have to be addressed. The paper is complex, there are data not always clear. The statistical analysis has not been performed appropriately and rigorously. The authors conclude that "[c]onsidering these clusters may reduce the uncontrolled differences in patients’ prognosis and improve the power of studies." It is not clear how the proposed clustering would help in clinical practice, authors should please clarify this point in the manuscript. 

There are no conflicts between the reviews.

In my personal opinion even if this is an interesting paper, there are major questions to answer, and I was in doubt with the really possibility to make it achievable. I'm asking you a complete major revision with a specific attention to the statistical analysis. 

Dear editor

We tried to solve the methodological points and to make the reading easier.

Indeed the new reviewers asked for more details and it is of course difficult to be easily accessible to common readers and to provide details to specialists in statistics.

We tried to provide details in ESM and to keep the manuscript understantable by common medical readers. 

Best regards

Prof JF Timsit MD PhD

Reviewer #1: In "Clinical and biological clusters of sepsis patients using hierarchical clustering." Papin et al. report on an analysis based on data from 6,046 sepsis/septic shock patients from the French national prospective multicenter ICU cohort OUTCOMEREA. Essentially, the authors performed a hierarchical clustering (HC) on a random sample of 4,050 patients and derived six clusters which differed in early or late mortality (even after adjustment on severity of organ dysfunctions (SOFA) and year of ICU admission). The authors conclude that "[c]onsidering these clusters may reduce the uncontrolled differences in patients’ prognosis and improve the power of studies." Addressing the heterogeneity of sepsis patients to contribute e.g. to more targeted randomized controlled trials, is very important. The authors use retrospective data from a registry covering the period between 1997-2015. However, even after extensive reviewing, several major questions remain that have to be addressed. 

#R1.1: First of all the paper lacks clarity with regard to the steps that have been conducted. In this regard figure S1 and S2 should be put into the main text. 

R.R1.1: We agree, to better understand the study design and the methods used, we added figure S1 (Schematic of study) into the main text. However, as Figure S2 give no crucial information; we think that it is better to leave it in the supplementary material. 

#R1.2: Afterwards, it has to be clear in all figures and tables if the training or the validation data set is displayed. 

R.R1.2: We agree, to improve understanding, all titles of figures or tables were completed by mention: “(performed in training set)” or “(performed in validation set)”. 

#R1.3: How many variables (overall) entered the HC – this number is important and is missing? 

R.R1.3: All 63 variables presented in Table 1 were included in the cluster analysis. To account for further comment, the Table 1 is now moved in appendix as S1 Table, as suggested in #R.R1.7 and missing values previously presented separately in the appendix were added to the S1 Table.

Moreover, to account for this comment understanding, the title of this table was modified as follow: ”S1 Table: Description of the 63 variables included in the cluster analysis”. Following sentences were added line 148: “Description of 63 variables included in the cluster analysis was available S1 Table” and the line 177: ”A MCA was performed to reduce the dimension of the 63 variables of the dataset (S1 Table) in “Euclidian patient-coordinates” dataset.”. 

#R1.4: If the HC is similarly conducted in the validation data set (independent of the training set) what cluster structure is derived and how similar is it to the six clusters? 

R.R1.4: We agree, more information on the clusters building in the independent validation set is needed. 

Regarding the similarities and differences between clusters identified on the training set and on the validation set, we have already provided distribution of all characteristics of cluster in validation set in S4 Table. As suggested in #R1.9 , the main analysis for assess similarity and reproducibility is given in the new S5 Table. 

Following information was added in the manuscript:” Six clusters were identified in the validation set according to the indices used to select the number of clusters (S9 Fig.). The contributions of variables in the construction of the first four dimensions of the MCA are depicted on S10 Fig. S4 Table described the distribution of demographic characteristics, various comorbidities, sources of infection, micro-organisms, clinical and biological data, and the organ failures at admission.”. 

Following information were added in the appendix: ”S9 Fig : Representation of indices used to determine the dendrogram cut level (performed in validation set).”. 

#R1.5: The focus should generally be on the validation data set! As this is the more critical step and of course generalizability to other ICU data sets is the ultimate step. Finally, and related to this point, the authors should extend the comparison to Seymour et al. (2019) and more systematically describe if and how their French results differ from those of the US electronic medical record analysis.

R.R1.5: We agree that external validation cohort is critical; it will be an upcoming study for our team. However, some external validation should be performed by using dataset with common collected variables, that is not the case. This information was available in the discussion section as follow: Second, the training set and the validation set come from the same database. A validation with an external database with common variables is needed.”

Other major points:

#R1.6: Page 8 (line 159-): What statistics are used for decisions and it is not clear why and how k-means clustering is included.

R.R1.6: At the best of our knowledge, there was no unique method or indicator neither consensus in the literature to define the optimal number of clusters. Here, we chose a trade-off between the four criteria used (Semi-partial R-squared, R-squared, Pseudo F statistic and pseudo t² statistics presented in S4 Fig.) and the clinical relevance of clusters (number of individuals by cluster and identity of the clusters). Following sentence was added in the method section: “There is no consensus in the literature on the final choice of the number of clusters regardless of the clustering method”. 

K-means algorithm was considered to identify potential outliers in the data. This information was added in the methods section. 

#R1.7: Table 1: Can be put to the Supplement – why did you categorize quantitative variables (which is not recommended!)

R.R1.7: We agree, the Table 1 was moved in appendix as S1 Table. Moreover, concerning the categorization of quantitative variables, as we used MCA, we need to have qualitative variables only, as mix type of variables cannot be considered here. Following sentence was added in the method section: “because MCA is based on qualitative variables, quantitative variables were categorized”.

#R1.8 : Page 10 (line 176-): These are time to event analyses (that can address censoring); thus, I hope that the mortality estimates are derived from the Kaplan Meier estimator (on page 11; line 206-); similarly I assume that Cox regression models were used for the analyses with adjustment?

R.R1.8: As specified (on page 11, line 206), it was crude mortality rates. As suggested, we replaced it with the estimated mortality by Kaplan Meier. Risks of mortality were estimated using Cox model. This information was added in the method section and in footnote as follow: “A cox model was used to determine the hazard ratio. Data are reported as HR ±95% confidence intervals, presented from the lowest to highest”

#R1.9: Page 10 (line 187): Accuracy in the training set is of very little use – report this information in the validation set, please.

R.R1.9: We agree, accuracy in the validation set is now provided in S5 Table. Accuracy in the training set remains available in S4 Table. 

#R1.10: Page 10 (line 188-): Especially the red new text is not clear; in particular what is meant by “new”?

R.R1.10: The term “new” has been deleted and methods section was modified as follow: “Cluster analysis was also applied in the validation set.”

#R1.11: Page 11 (line 210-): What is meant by “The clusters supported 12% of the total variability.”?

R.R1.11: The term “variability” refers to the observed differences between all individuals of the dataset. Due to the poor information provided, this sentence was deleted.

#R1.12: Page 20 (line 333-): Is also not clear (I refer to the other points above).

R.R1.12: As specified in #R.R1.4, sentence was modified as follow: “Six clusters were identified in the validation set according to the indices used to select the number of clusters (S9 Fig.). The contributions of variables in the construction of the first four dimensions of the MCA are depicted on S10 Fig. S4 Table described the distribution of demographic characteristics, various comorbidities, sources of infection, micro-organisms, clinical and biological data, and the organ failures at admission.”..

#R1.13 : Page 21 (line 358-): This recommendation should be attenuated and written more carefully.

R.R1.13: The paragraph was modified as follow: “We identified a small cluster that included patients with COPD exacerbation. These patients met to the definition of sepsis 3.0, yet mechanisms of organ failure were probably very different from other cluster. The inclusion of these patients in sepsis studies should be discussed.”

#R1.14 : Page 21 (line 364-): Please shorten the limitations part and structure the arguments better

R.R1.14: We agree, limitations section was shortened and better stratified (#R.R1.16).

#R1.15 : Page 22 (line 391-): The argument regarding “selection bias” is not clear.

R.R1.15: As specified in #R.R1.14, the sentence was deleted. 

#R1.16 : Page 23 (line 403-): “Second, …” where is first?

R.R1.16: We agree, as specified in #R.R1.14.

Minor points:

#R1.17 : Page 5: The authors should mention the period of data collection already; moreover did the 20 ICUs contribute homogeneously along these 18 years? Please provide information on the ICUs their focus and temporal contribution

R.R1.17: Period of data collection was already mentioned line 205 and discussed line 358. Moreover, the 20 ICUs contributed heterogeneously. The S7 table was added in the appendix and the following sentence was modified in the discussion section as follow: “Third, the period of patient’s admission on our study ranged from 1997 to 2015 with heterogeneous periods of inclusion and number of patients included between each center (S7 Table)”. 

#R1.18 : Page 7 (line 141-): Please provide a little more information on the reported coefficients

R.R1.18: The term « κ -coefficients » was modified by Cohen's kappa coefficient.

#R1.19 : Page 7 (line 146-): Please provide more information on the missing value handling including the imputation and most importantly the impact that the imputation has on the conclusions

R.R1.16: As specified in #R.R1.3, missing values was presented to the S1 Table.

#R1.20 : Page 23 (line 414): better use prognosis instead of “prognostication” throughout

R.R1.20: The term “prognostication” was replaced by “prognostic estimation”.

 

Reviewer #2: Dear editor,

thank you very much for inviting me to review this paper.

The paper is a complex manuscript about the necessity to divide sepsis patients in cluster based on disease characteristics. Authors individuated six different clusters of patients analyzing data from a multicenter prospective database.

#R2.1: Even though I lack the expertise to judge the details of the methodology, the chosen approach to explain data is not clear and the deep use of figures and tables makes all the results incomprehensible. I would suggest explaining the results, figures, and tables in text to allow a better understanding.

R.R2.1: The unsupervised clustering we used is only devoted to reduce heterogeneity between individuals by emphasizing resemblance at the ICU admission. The originality of this technique is that it does not consider any specific outcome. It is like sorting people in a crowd based on whether they are tall or not, because this specification is the best way to dichotomize this crowd into two more homogeneous samples. Interestingly, in our study, these specific clusters, defined based on patient’s characteristics at admission, were associated with different susceptibility to therapy but also different health care consumption, duration of stay and mortality.

#R2.2: Indeed, it is unclear what kind of information this study would add to daily clinical practice. It is clearly evident how sepsis and septic shock are conditions deeply related to clinical characteristics of the host, microbes involved and concomitant diseases, but it is not clear how the proposed clustering would help in clinical practice, I would suggest to better clarify this point in paper.

R.R2.2: The six identified clusters reduced the heterogeneity of the syndrome and allowed a better discrimination of patients with sepsis at ICU admission based on characteristics available at admission. These clusters are characterized by distinct pathophysiological signatures, and different mortality risks and resources consumption. Interestingly, as already highlighted in the manuscript, the severity of illness and of organ dysfunctions are not the main sources of heterogeneity in septic population. 

The following sentence was added in the discussion section line 358:” The ability to affect a patient with sepsis to a homogeneous cluster should enable to achieve a personalized ICU medical care strategy, and to test potentially appropriate new therapies by performing more efficiently targeted clinical trials. The simplest way to do so is to use a classification tree.”

#R2.3: Language is correct and all the bias and limits of the study are well indicated in manuscript.

---

## [Editor Report · Decision Letter 1]

14 May 2021

PONE-D-20-35126R1

Clinical and biological clusters of sepsis patients using hierarchical clustering.

PLOS ONE

Dear Dr. Papin,

Thank you for submitting your manuscript to PLOS ONE. After careful consideration, we feel that it has merit but does not fully meet PLOS ONE’s publication criteria as it currently stands. Therefore, we invite you to submit a revised version of the manuscript that addresses the points raised during the review process.

 ACADEMIC EDITOR:  

Dear authors,

thanks for your answers.

The aim of the study is clearly to identify clusters of patients based on characteristic available at patients’ admission. My concerns is about clinical involvements. If your aim is to propose a different clinical strategy, you should better clarify this point. It is clear that different clusters have different clinical outcomes, health care consumption, duration of stay and mortality; anyway, it is not clear why an ICU physician should use this classification to plan clinical strategy. Thanks for the sentence you added to line 358, but I would specify that further studies are necessary to better explain this clustering. 

We look forward to receiving your revised manuscript.

Kind regards,

Martina Crivellari

Academic Editor

PLOS ONE
---

## [Author Response · Author response to Decision Letter 1]

20 May 2021

ACADEMIC EDITOR: 

Dear authors,

Thanks for your answers.

#AE1: The aim of the study is clearly to identify clusters of patients based on characteristic available at patients’ admission. My concerns is about clinical involvements. If your aim is to propose a different clinical strategy, you should better clarify this point. 

R.AE1: We agree, the aim of the study is clearly to identify clusters of patients based on characteristic available at patients’ admission. This sentence was added in the abstract (line 39) as follows: “The aim of the study is to identify clusters of patients based on clinical and biological characteristic available at patients’ admission.” and in the methods section (line 99) as follows: “The primary objective of our study was to identify clusters among patients with sepsis by considering data available at admission including: underlying disorders, source of infection, micro-organism, biological host response and organ failures.”. 

#AE2: It is clear that different clusters have different clinical outcomes, health care consumption, duration of stay and mortality; anyway, it is not clear why an ICU physician should use this classification to plan clinical strategy. 

R.AE2: To clarify the interest of the clustering of sepsis patient’s the sentence of the introduction section (line 77) was reformulated as follows: “As explained by J. Castela Forte, to personalize and improve treatments of sepsis, patients must be clustered into common phenotypes based on clinically objective parameters reflecting disease mechanisms. After external validation step, these clusters could be grouped by underlying causal mechanisms and will improve patient characterization; optimized design and powering of randomized control trials. Finally, these clusters can allow identifying differential response patterns by considering baseline characteristics of sepsis patients.”

#AE3: Thanks for the sentence you added to line 358, but I would specify that further studies are necessary to better explain this clustering.

R.AE3: We agree, sentence was modified as follow : ”First, these phenotypes will reduce heterogeneity prior to randomization; however, because they will not necessarily create better trials, further studies are necessary to better explain this clustering”.

---

## [Editor Report · Decision Letter 2]

24 May 2021

Clinical and biological clusters of sepsis patients using hierarchical clustering.

PONE-D-20-35126R2

Dear Dr.Papin,

We’re pleased to inform you that your manuscript has been judged scientifically suitable for publication and will be formally accepted for publication once it meets all outstanding technical requirements.

Kind regards,

Martina Crivellari

Academic Editor

PLOS ONE
---

## [Editor Report · Acceptance letter]

23 Jul 2021

PONE-D-20-35126R2 

Clinical and biological clusters of sepsis patients using hierarchical clustering. 

Dear Dr. Papin:

I'm pleased to inform you that your manuscript has been deemed suitable for publication in PLOS ONE. Congratulations! Your manuscript is now with our production department. 

Kind regards, 

on behalf of

Dr. Martina Crivellari 

Academic Editor

PLOS ONE